# Owner-Reported Health Events in Australian Farm Working Dogs

**DOI:** 10.3390/ani14131895

**Published:** 2024-06-27

**Authors:** Nicola Ann Pattison, Katja Elisabeth Isaksen, Naomi Cogger

**Affiliations:** School of Veterinary Science—Tāwharau Ora, Massey University, Private Bag 11 222, Palmerston North 4442, New Zealand; nic_pattison@hotmail.com (N.A.P.); n.cogger@massey.ac.nz (N.C.)

**Keywords:** survey, health, Kelpie, prevalence

## Abstract

**Simple Summary:**

Working farm dogs are very important to livestock farming in Australia, but no studies have investigated their overall health. This study aimed to determine which diseases and injuries are most common in Australian working farm dogs. A total of 125 farm dog owners were recruited at a public working dog event and interviewed about the health of their dogs in the preceding 12 months. Information was provided about 526 dogs. Most dogs were Kelpies and were not spayed or neutered. Approximately 9% of the dogs had been sold, 6% had died, and 1% had been retired from work. About one-quarter of the 444 dogs under the respondents’ care had an illness or injury in the previous 12 months. Most injuries were fractures, tendon or joint injuries, or skin wounds, and most dogs with an illness had arthritis. A little less than half of the injured dogs and a quarter of the ill ones were treated by a veterinarian. The results suggest that they are similar to other working farm dogs. However, Australia is unique, and further research is needed to ensure the best care for farm dogs here.

**Abstract:**

Working farm dogs are an integral part of livestock farming in Australia but, currently, no studies have investigated their overall health. This study aimed to identify the prevalent diseases and injuries among these dogs, offering a baseline dataset. A total of 125 farm dog owners were recruited at a public working dog event and interviewed about the health of their dogs in the preceding 12 months. Information was provided about 526 dogs. Most dogs were Kelpies, and not neutered. Nine percent of the dogs had been sold, 6% had died, and 1% had been retired. Of the 444 remaining dogs, 24% had a health event in the past 12 months, while 13% had a health event that affected their work. The injuries (11%) mostly involved the musculoskeletal system or skin, while the non-traumatic illnesses (13%) commonly included arthritis. Forty-five percent of the injured dogs and 26% of the ill ones received veterinary treatment. The results indicate similarities to previous studies into working farm dogs. Given their unique challenges, further research is needed to ensure optimal care for farm dogs in Australia.

## 1. Introduction

There are around 33,500 broadacre livestock farms in Australia [1], and working farm dogs are an integral part of many farming enterprises [2]. The work these dogs undertake is exceptionally demanding, with dogs travelling 20 to 30 kilometres per day on average [3,4] and in ambient temperatures of as high as 30 °C [3]. To work effectively in such conditions, dogs must be fit and healthy. This is underscored by a survey in which nearly all dog handlers in the private sector, including farmers, rated health as important or very important to the welfare of their dogs [5]. However, while studies aimed at describing the population, behaviours, management, training and genetics of working farm dogs in Australia have been carried out [2,4,5,6,7,8,9,10,11,12,13,14,15], few have focused on dogs’ overall health.

Several studies have examined the health of working farm dogs in New Zealand. Here, farm dogs have been found to have a median body condition score of four out of nine [16], with very few dogs being reported as overweight [16,17]. Traumatic injuries, usually affecting the skin and limbs, were found to be common and were often reported to have been caused by vehicles, crossing of fence lines, or encounters with livestock [17,18]. Musculoskeletal diseases and abnormalities were reported to be common when surveying owners [17,19], analysing veterinary hospital records [18], and examining dogs on farms [16,20], and a longitudinal study reported that lameness may double the dogs’ risk of being lost from the workforce [21]. These New Zealand studies can provide useful comparisons to Australian dogs. However, due to substantial differences in the environments, farming practices, and which dog breeds are used to work with livestock, the results of studies conducted in New Zealand may not be transferrable to Australian working farm dogs. For example, in Australia, one survey reported that almost 70% of herding dogs were Kelpies or Kelpie crosses, and about a quarter were Border Collies or Border Collie crosses [5], while in New Zealand, almost all were Heading dogs or Huntaways [16,19,22]. Additionally, the climates of the two countries differ substantially. Australia has a huge variety of weather patterns including large areas of arid grassland and desert [23], while New Zealand is largely temperate with subtropical areas in the far north [24]. These climate differences impact the weather conditions that the dogs work in and how the land is farmed, with Australian dogs likely being exposed to higher temperatures and more arid conditions than New Zealand dogs.

This paper presents the results of a cross-sectional study of owner-reported health information for Australian working farm dogs. The study had three aims: First, to describe the preventative health care practices of farm dog owners; second, to estimate the prevalence of health events; and third, to investigate the reasons for dogs being euthanised, dying, or being retired from work.

## 2. Materials and Methods

### 2.1. Study Design

A survey was conducted between 6 June and 8 June 2015 during the annual Kelpie Muster event in Casterton, Victoria, Australia. The survey was designed to collect data about the population characteristics, husbandry and management, adverse health events, and causes of death or euthanasia in working farm dogs in Australia. Data were collected through structured face-to-face interviews with owners of working farm dogs who were recruited at a major event celebrating Kelpies, a common and iconic breed of working farm dog found in Australia.

### 2.2. Recruitment of Working Farm Dog Owners

Working farm dog owners were recruited to participate in a cross-sectional study at the 2015 Kelpie Muster in Casterton, Victoria, Australia. The Kelpie Muster is an annual event celebrating Australian working farm dogs. We aimed to interview a minimum of 100 dog owners to enable us to collect data on at least 384 dogs. The required number of dogs was based on an expected prevalence of 50% for the different conditions, a precision of +/−5% and a confidence level of 95. Deviations from 50% in either direction would have reduced the number of dogs required. A minimum of 100 dog owners was selected because previous surveys have found that farm dog owners have approximately three to five working dogs each [2,25].

During the 2015 Kelpie Muster, a stand was set up near the entrance to the event. People were asked to participate when they approached the stand or were approached by a veterinarian (NP) or a veterinary student who were both circulating throughout the event. If a person agreed to participate, they were included in the study if they had owned one or more working farm dogs in the year preceding the survey and used those dogs to work livestock corresponding to at least 100 Dry Sheep Equivalents. The dogs had to be at least 12 months old on the day of the survey or on the last day that it was on the dog owner’s property.

Dry Sheep Equivalents are a unit of measurement used in Australia to calculate feed requirements for different types of livestock [26]. One Dry Sheep Equivalent is defined as corresponding to one two-year-old, 45 kg merino sheep (castrated, non-pregnant, or non-lactating).

### 2.3. Data Collection

If a dog owner met the criteria and agreed to participate, data were collected through a structured interview using a questionnaire (see Appendix A). A veterinarian (NP) read the questions to participants and recorded their answers. Each interview took approximately 15 min to complete. The questionnaire was pre-tested by people knowledgeable about working farm dogs in Australia, and necessary adjustments were made.

The final version of the questionnaire had three sections. The first section collected data about the farm dog owner, the farming operation they worked on, and the general preventative health care they routinely administered to their working dogs.

The second section collected data about the individual working farm dogs that the dog owners had under their care at the time of the survey, who were older than 12 months. For each dog, information was collected about age, sex, breed, type of work performed, type of livestock worked, and history of disease and injury in the previous year. The type of work the dogs performed was recorded as “paddock”, “yard”, or “all-rounder”. Dogs’ work was classified as “paddock” if their work primarily involved mustering and moving stock between paddocks and into yards. Yard dogs were specialised dogs that worked with stock in the yards or moving stock onto trucks. All-rounders were those who worked in all these environments.

The third section of the questionnaire collected information about working farm dogs that had left the participant’s care in the previous year.

To be included in the study, dogs need to be at least one year old on their last day under the care of the respondent. For eligible dogs, the age, breed, sex, and reason the dog was no longer under the participant’s care were recorded. If the dog had died, the cause of death, whether the dog was euthanised or died naturally, and the person who performed the euthanasia were recorded.

### 2.4. Classification of Adverse Health Events

Recorded health events were classified as traumatic injuries or non-traumatic illnesses. A disease event was classified as traumatic if it was known to be the direct result of a traumatic incident, and illness if no traumatic incident was identified. Traumatic injuries were further categorised according to the type, anatomical location, and cause of the injury (Table 1). If the dog received veterinary treatment for a traumatic injury, the treatment type was classified as medication, surgery, hospitalisation, or other. Non-traumatic illnesses were categorised as degenerative or age related, genitourinary, skin, ear or eye, or other. A record was made of whether dogs had received veterinary treatment for a non-traumatic illness. In classifying the health information, the category “other” was used to group events that occurred in fewer than five dogs.

### 2.5. Data Analysis

Data were analysed using Microsoft Excel 365 and R version 4.3.3 [27]. As we are presenting data for descriptive purposes, no significance testing was carried out.

When a dog experienced multiple health events, the data collection process did not provide sufficient information to determine whether these events’ types, locations, causes, or treatments were related or stemmed from separate occurrences. As a result, the study quantified the number of individual dogs affected by each type of health event but did not attempt to determine the total number of separate health events experienced by each dog.

## 3. Results

### 3.1. Dog Owners

The study population comprised of 125 dog owners, with 123 who currently owned dogs and 2 who did not currently own a dog but had in the previous 12 months. Most dog owners were male, older than 40 years of age, farm owners, and lived in Victoria (Table 2). The median size of the property that dog owners owned or worked on was 809 hectares (n = 114 dog owners, range = 25–30,000 hectares). The 123 dog owners who owned dogs at the time of the survey had a median of three working farm dogs each (range = 1–13 dogs).

Of the 125 dog owners, 109 (87%, 95% CI = 81–93%) had the same protocols for vaccination, deworming, and external parasite control for all their dogs. A further 16 (13%, 95% CI = 7–19%) used different routines for each of their dogs. Individual data on the treatment of these dogs were not collected. The treatment regimens used by the 109 dog owners are summarised in Table 3.

### 3.2. Dog Population

#### 3.2.1. All Dogs

Data were collected on a total of 526 working farm dogs in Australia. The median age of dogs was 3 years (range = 1–16). The most common breeds were Kelpies or Kelpie crosses (n = 463), followed by Border Collies (n = 39), and the remaining 24 dogs were other breeds or unknown. Among the 526 dogs, 84 (16%) were no longer under the care of their owners. Forty-six dogs (9%, 95% CI = 6–11%) had been sold, 33 (6%, 95% CI = 4–8%) had died, and 3 (1%, 95% CI = 0–1%) had been retired. The median age at death was 8 years, with a range of 1 to 15 years. The most frequently reported cause of death or euthanasia during the twelve-month study period was an illness (Table 4).

In total, 59 dogs (11%) were neutered (Table 5), 39 of which (72%, 95% CI = 60–84%) were reported to have been neutered for management reasons, which included a wish to prevent unwanted litters or to reduce aggression. A total of 10 dogs (19%, 95% CI = 8–29%) had been neutered for health reasons, which included reported vaginal prolapse (5 dogs), conditions relating to pregnancy (3 dogs), and prostate cancer or infection (2 dogs). Ten dogs did not have a reported reason for having been neutered.

#### 3.2.2. Dogs in the Owner’s Care at the Time of the Survey

Four hundred forty-four dogs were still in the owners’ care on the day of the survey. Of these, 267 (60%) were registered with the local shire or council, and 54 (12%) were insured. Data on past or future breeding plans were available for 387 entire dogs that were in the owners’ care at the time of the survey. A total of 156 (40%) had been used for breeding in the past or were planned to be used for breeding in the future. Forty-seven percent (85 of 181, 95% CI = 40–54%) of the entire females and 35% (71 of 206, 95% CI = 28–41%) of the entire males had been or were planned to be used for breeding.

Of the 444 working farm dogs still under the owners’ care at the time of the survey, 412 (93%, 95% CI = 90–95%) were reported to have worked with stock recently. The majority of these dogs did not compete in dog herding trials, tended to work in stock yards and paddocks, and worked either with both sheep and cattle or with sheep only (Table 6). Types of stock classified as “Other” included horses (3 dogs) and ducks (1 dog). Of the 32 dogs not working with stock, 15 were retired from work, 2 were not working due to illness or injury, 1 was too young, 1 was kept as a pet due to not being trained at a young age, and 13 had no recorded reason for not working.

### 3.3. Health Events

Detailed health data were available for the 444 working farm dogs under the owners’ care on the day of the survey. Of these dogs, 106 (24%) were reported to have experienced one or more health events in the 12 months before the survey. Of the 106 dogs with a health event, owners reported that the work of 59 dogs (56%) had been affected. Furthermore, 21 of the 59 dogs (26%) had not returned to full work at the time of the survey, including 8 (14%) dogs that had been retired following the health event.

Traumatic injuries were recorded in 53 of 444 dogs (12%) (Table 7), where 45 (85%) of the 53 injured dogs received veterinary treatment, 35 (66%) were admitted to a veterinary hospital, and 21 dogs (40%) were recorded to have received surgical treatment. The most commonly recorded reason for injuries were accidents involving vehicles (Table 7). For the 16 dogs (of 19) where information was available, all these accidents occurred when dogs fell or jumped off moving vehicles.

In addition, 57 of the 444 dogs (13%) were reported to have been affected by a non-traumatic illness during the study period (Table 8). Fifteen of the 57 dogs (26%) with an illness were seen by a veterinarian. Of the 18 dogs with a degenerative illness, 13 were noted to have arthritis, stiffness, or difficulty moving.

Types of illnesses classified as “Other” included heat stress and infections (4 dogs each); genetic illness, conditions affecting the limbs or conditions affecting the oral cavity (3 dogs each); cancer, parasitic infections, and conditions affecting the nervous system and spine (2 dogs each); and poisoning and conditions relating to the gastrointestinal, immune or respiratory/circulatory systems (1 dog each).

## 4. Discussion

This paper presented population and health data of 526 Australian working farm dogs that had been in the respondents’ care in the previous 12 months. A majority of the dogs were Kelpies or Kelpie crosses, and the majority were sexually entire. We found that nearly a quarter of the dogs had experienced at least one health event in the 12 months before the survey, with traumatic injuries and non-traumatic illnesses each affecting a little over 1 in 10 dogs. The most common types of injuries were lacerations or skin grazes, fractures, and tendon or ligament injuries. In contrast, degenerative conditions, genitourinary, and skin problems were the most frequently reported illnesses. Approximately 15% of the dogs were no longer under their owners’ care, with the majority having been sold, about 6% of dogs having died, and only three (1%) having been retired from work.

The median age at death for working farm dogs in this study was 8 years. This is consistent with results from adult New Zealand farm dogs, where the most common age at death was 7 to 9.9 years [21]. Military dogs and guide dogs have been reported to live somewhat longer, at an average of around 10 to 11 years [28,29]. However, it is difficult to directly compare our results with those of studies conducted with non-farming working dogs. Such dogs are probably more likely to be retired rather than euthanised, as there are usually clear procedures for when they should be retired from duty and what should happen to them after retirement. In comparison, decisions about retiring, re-homing, and euthanising working farm dogs are the sole responsibility of their owner and depend on their individual needs and resources, and their perception of the dog’s needs. Given that farm dogs need high levels of activity, it may be difficult to re-home them as pets. Additionally, many farm dog owners are likely to live far from veterinary clinics, making it difficult for them to seek prompt treatment for sick or injured dogs, especially in acute, life-threatening cases. For these reasons, working farm dogs are probably more likely than other working dogs to die or be euthanised when they are seriously ill or injured, or when they can no longer work. This is supported by studies of US military dogs [30], New Zealand police dogs [31] and British guide dogs [28] that reported retirement ages of around seven to nine years, which is similar to the reported ages at death in farm dogs.

In this study, only three dogs were reported to have been retired. This is lower than that reported in a New Zealand study [21], but as the data cover a much shorter time period, it is likely to be comparable. In addition to the working farm dogs being more likely to be euthanised than retired, the low number of retired working farm dogs in these studies may be a result of retirement not being clearly defined for these dogs. A proportion of older dogs in this study may have had their work reduced or been partially retired but are still considered to be “in work”, causing us to somewhat underestimate the number of retired dogs.

Overall, about one in four dogs in this study were reported to have experienced one or more adverse health events in the previous 12 months, with 12% of dogs experiencing an injury and 13% experiencing an illness. These proportions are much lower than those reported by Sheard, who found that 52% of working farm dogs in New Zealand had a non-traumatic and 25% had a traumatic health event in a twelve-month period [17]. Differences in how the data were collected can explain most of these differences, as Sheard collected data through pre-arranged interviews with dog owners at home. In the current study, participants were recruited and interviewed at a busy event. They did not have time to refresh their memories beforehand and were likely to have other plans for the day, making it more likely that they would forget to report some health events. This is especially likely to have affected the reporting of less serious events, such as minor infections or injuries that did not affect the dogs’ ability to work. Additionally, some conditions may go unnoticed by owners until they are more advanced, simply because owners are not always aware of subtle clinical signs or symptoms. For these reasons, it is expected that the prevalence of adverse health events in this study is likely to have been underestimated. While this underestimation may have been avoided by clinically examining every dog included in the survey, this would not have been possible due to the nature and time frame of the study.

The most common types of traumatic injuries were motor vehicle injuries or injuries caused by stock, and they commonly affected the musculoskeletal system and skin. These results are generally consistent with those reported by other studies of working farm dogs [16,17,18] and are likely to be a reflection of the risks involved when dogs work with livestock in a farming environment. Due to the nature of the dogs’ work, it is likely difficult to completely prevent stock- and vehicle-related injuries. Some interventions, such as removing the gaps in rails or bars that dogs can get caught on or ensuring that dogs stay still while vehicles are moving, may help prevent some accidents. However, as we do not have any specific information about the nature of these incidents, recommendations are difficult to make.

Musculoskeletal injuries and degenerative illnesses were the most commonly reported conditions in this study, and the majority of degenerative illnesses were noted to affect the musculoskeletal system. Musculoskeletal disease has been found to be the most common cause of retirement or euthanasia in guide dogs [28], police dogs [31] and military dogs [29,30], while lameness doubled the risk of working farm dogs dying or being retired in a six-month period [21]. Injuries can lead to diseases such as osteoarthritis later in life [32]. Therefore, preventing injuries can benefit both working dogs and their owners, as it improves the dogs’ long-term health and welfare and is likely to increase their working lifespans. However, more information is needed about factors that might put dogs at an increased risk of injury or musculoskeletal disease, and how these might be avoided. Future research into the health of working farm dogs should examine such risk factors and aim to suggest ways in which musculoskeletal injury and disease might be avoided.

Further studies are needed to determine the types of injuries and diseases commonly affecting working farm dogs in Australia, and how these may be prevented. Ideally, longitudinal studies would be conducted to enable researchers to make robust conclusions about risk factors related to outcomes such as chronic musculoskeletal disease and dogs being lost from the workforce. Additionally, if health and welfare outcomes are to be improved for working farm dogs, researchers are dependent on the cooperation of people within the livestock production industry to implement any recommendations. Therefore, it would also be useful to investigate the reasons why farmers may be prevented from following best practices regarding dog husbandry or seeking veterinary care and treatment for their dogs.

By far, the most common breed in our study were Kelpies (84%). This is consistent with a previous survey of Australian farm dogs by Arnott et al. [2]. However, it is possible that the proportion of Kelpies in both studies may have been inflated due to the way study participants were recruited. In our study, participants were recruited at an event focusing on Kelpies, while Arnott et al. recruited dog owners partially by advertising on the website of the Working Kelpie Council of Australia. However, events and organisations focusing solely on working Kelpies are likely to exist and be successful due to the fact that Kelpies are a dominant breed in the Australian working farm dog population. In comparison, very few Kelpies have been found to be working stock in New Zealand [16,17,18,19,22].

There were an approximately equal number of male and female dogs, and a majority of the dogs were not desexed. Our results are comparable to previous Australian and New Zealand studies, which have all found that most working farm dogs are sexually entire [2,16,17,19,33]. Similar costs and benefits of neutering dogs probably exist for Australian farmers as they do in New Zealand, where it has been speculated that they may want to be able to breed dogs that turn out to be good workers, while the cost in terms of money and recovery time may be considered to outweigh the benefits [16]. This may cause many farm dog owners to only neuter dogs when they have a specific reason to do so, such as a wish to avoid mismating or reduce fighting in their team, or where the dog develops a health problem related to its reproductive system.

Beyond removing the risk of pregnancy and injury or disease of the reproductive system, such as pyometra and testicular injury, the evidence for whether or not desexing is beneficial for dogs is mixed. One study found that neutered dogs live slightly longer, but the authors caution against over-interpreting their results due to the way the data were collected and analysed [34]. Depending on the dog’s breed and age at neutering, desexing may increase the risk of dogs developing female urinary incontinence [35,36], while the risk of certain joint disorders and some cancers increase, and the risk of other cancers decrease [37,38,39]. When considering that musculoskeletal disease and injury are common in farm dogs [17,18,20] and lameness is a major risk factor for dogs dying or being retired [21], it is concerning that neutering may be associated with an increase in joint disease. However, a systematic review found that desexing is associated with reductions in aggressive behaviour and the risk of dogs biting humans [40], meaning that desexing may reduce aggression and the risk of bite injuries. Given the mixed evidence, more information is needed to determine how the low rate of desexing affects the health and welfare of working farm dogs.

Almost three of five farm dog owners vaccinated their dogs yearly, and the same number administered prophylactic deworming treatments every one to three months. Few farmers reported gastrointestinal, infectious, or parasite-related diseases in their dogs, suggesting that the current practices may be effective at managing these diseases in working farm dogs. However, it is also possible that some cases went unnoticed by dog owners and were therefore not reported. A 2014 study found that about 30% of 1425 rural dogs in Eastern Mainland Australia and Northern Tasmania were infected by helminths, and that farmers relied mainly on anthelmintic products to manage the problem [41]. While that study did not report the proportion of dogs that showed clinical signs of disease, it does suggest that more could be done to eliminate internal parasite infections in rural dogs in Australia.

While this paper provides a valuable first look at the health of Australian farm dogs, it has a number of limitations. Firstly, as the data are based on retrospective owner reports rather than veterinary examinations, there is a risk of under-reporting of less impactful health events such as minor injuries or diseases that did not require treatment. Secondly, as the data were collected during an event focused on Kelpies, there may have been an over-estimation of the number of Kelpies in the population. Third, due to a lack of detail in the recording of health events, we were unable to determine how many health events the dogs had experienced, or link the types of injuries/diseases, body parts affected, and causes to specific events. For example, we were able to report how many dogs experienced broken bones and how many experienced motor vehicle injuries, but we were not able to report how many dogs experienced broken bones caused by motor vehicle injuries. Due to these limitations, we chose to present this data in a descriptive manner rather than attempting any risk factor analysis. Future studies with more rigid data collection methods may be able to build on our results and conduct more in-depth analyses.

In addition to the limitations around data collection and analysis, the time that passed between data collection and publication may be a cause for concern. Global events such as the COVID-19 pandemic and following economic impacts may have affected some farmers’ ability to provide optimal care for their working dogs. However, while this may be a welfare concern in relation to the cost and availability of, for example, veterinary treatments, it seems unlikely that farming practices will have changed enough to affect the overall population and work of dogs on Australian farms to the point where the results presented here are no longer valid. A greater concern may be the rapidly changing climate of Australia, which is resulting in higher frequencies of extreme heat, droughts, bush fires, and rainfall [42]. These changes may impact the health and welfare of working farm dogs and should be considered in future studies.

## 5. Conclusions

This study aimed to identify and describe the most common types of diseases and injuries in the Australian working farm dog population. Such data are vital to guide further research into conditions that strongly impact the dogs’ health and welfare. Identifying the most common diseases will allow more targeted research to identify the risk factors that may determine the likelihood of dogs suffering adverse health events, thus allowing researchers to make specific recommendations to improve health and welfare outcomes in this group of dogs.

The results presented provide a baseline dataset for working farm dogs Australia and suggests that this population is largely similar to previously studied dogs, notably working farm dogs in New Zealand. However, Australia has unique environmental conditions, farming practices, and dogs breeds used for livestock work. Further studies are necessary to ensure that working farm dogs in Australia are provided with the best possible care.

## Figures and Tables

**Table 1 animals-14-01895-t001:** Categorisation of traumatic injuries in dogs.

Injury Category	Traumatic Injuries
Types	Lacerations/wounds, bone fractures, joint dislocation or injuries, tendon or ligament ruptures, puncture wounds, internal injuries, burns, or other.
Anatomical Locations	Hindleg, foreleg, hip, paw, head, ear or eye, abdomen, spine, chest, tail, or other.
Causes	Vehicle accidents, livestock, dog fights or bites, transit of fences, or other.

**Table 2 animals-14-01895-t002:** Demographic data relating to 125 dog owners who participated in the survey.

Variables		Number of Dog Owners	% (95% CI)
Sex	Female	25	20 (13–27)
	Male	97	78 (70–85)
	Not reported	3	2 (0–5)
Age range	20 or younger	3	2 (0–5)
	21–30	21	17 (10–23)
	31–40	27	22 (14–29)
	41–50	29	23 (16–31)
	51–60	26	21 (14–28)
	61–70	14	11 (6–17)
	Older than 70	5	4 (1–7)
Job title	Farm owner	86	69 (61–77)
	Employee	18	14 (8–21)
	Manager	12	10 (4–15)
	Contractor	8	6 (2–11)
	Not reported	1	1 (0–2)
Home state	Victoria	81	65 (56–73)
	South Australia	32	26 (18–33)
	New South Wales	8	6 (2–11)
	Tasmania	2	2 (0–4)
	Queensland	1	1 (0–2)
	Western Australia	1	1 (0–2)

**Table 3 animals-14-01895-t003:** The frequencies of preventative treatments given to working farm dogs by 109 dog owners who had identical treatment regimens for all their working dogs.

Treatments	Frequency	Number of Owners	% (95% CI)
Vaccination	Yearly	67	61 (52–71)
	Every two to three years	4	4 (0–7)
	Only as a pup	19	17 (10–25)
	Sporadically	13	12 (6–18)
	Never	6	6 (1–10)
Worming	Every one to three months	67	61 (52–71)
	Every four to six months	16	15 (8–21)
	Yearly	14	13 (7–19)
	Sporadically	11	10 (4–16)
	Unknown	1	1 (0–3)
Flea or mite treatment	Every one to two months	4	4 (0–7)
	Every three to six months	8	7 (2–12)
	Yearly	1	1 (0–3)
	Sporadically	42	39 (29–48)
	Never	27	25 (17–33)
	Unknown	27	25 (17–33)

**Table 4 animals-14-01895-t004:** Reported reason of death and median ages at death (in years) of 33 Australian farm dogs that died in a twelve-month period.

Reason for Death	Number of Dogs (%)	Median Age (IQR)
Illness	14 (42%)	8 (3–12)
Injury	8 (24%)	6 (2–10)
Old age	7 (21%)	14 (13–14)
Behaviour	3 (9%)	1 (1–2)
Not reported	1 (3%)	14 (-)
All dead dogs	33 (100%)	8 (2–13)

**Table 5 animals-14-01895-t005:** Sex and neuter status of 526 working farm dogs in Australia.

Neuter Status	Sex	Number of Dogs	% (95% CI)
Entire	Female	207	39 (35–44)
	Male	244	46 (42–51)
	All entire	451	86 (83–89)
Neutered	Female	35	7 (5–9)
	Male	24	5 (3–6)
	All neutered	59	11 (9–14)
Not recorded	Not recorded	16	3 (2–5)

**Table 6 animals-14-01895-t006:** Details about the types of work performed in the previous 12 months by 412 Australian working farm dogs reported to have been working with stock recently.

Variables		Number of Dogs	% (95% CI)
Herding dog trials	Casual	47	11 (8–14)
	Competitive	36	9 (6–11)
	Both	4	1 (0–2)
	Does not compete	322	78 (74–82)
	Not recorded	3	1 (0–2)
Type of work	Yard	370	90 (87–93)
	Paddock	365	89 (86–92)
	Truck	26	6 (4–9)
	Guard	4	1 (0–2)
	Not recorded	13	3 (1–5)
Combinations of work	Paddock and yard	323	78 (74–82)
	Paddock only	27	7 (4–9)
	Yard only	21	5 (3–7)
	Paddock, yard and truck	15	4 (2–5)
	Yard and truck	11	3 (1–4)
	Guard only	2	0 (0–1)
	Not recorded	13	3 (1–5)
Types of stock	Sheep	398	97 (95–98)
	Beef cattle	230	56 (51–61)
	Goats	25	6 (4–8)
	Dairy cattle	20	5 (3–7)
	Other	4	1 (0–2)
	Not recorded	1	0 (0–1)
Combinations of stock ^a^	Sheep and cattle	202	49 (44–54)
	Sheep only	170	41 (37–46)
	Sheep, cattle and other	17	4 (2–6)
	Cattle only	11	3 (1–4)
	Sheep and other	9	2 (1–4)
	Cattle and other	2	0 (0–1)
	Not recorded	1	0 (0–1)

^a^ Beef and dairy cattle are combined as “Cattle”. Goats are included in “Other”.

**Table 7 animals-14-01895-t007:** Traumatic injuries reported in 444 Australian working farm dogs in a twelve-month period. The types, anatomical locations, and causes of injuries are shown.

Category	Variable	Number of Dogs	Percentage
Type of injury	Laceration or skin graze	16	4%
	Fracture	10	2%
	Tendon or ligament injury	9	2%
	Joint dislocation or injury	8	2%
	Puncture wound or penetrating injury	6	1%
	Other	10	2%
	Not recorded	3	1%
Anatomical location	Hindleg	15	3%
	Foreleg	12	3%
	Paw	11	2%
	Hip	9	2%
	Head	5	1%
	Other	14	3%
	Not recorded	2	<0.5%
Cause of injury	Motor vehicle	19	4%
	Injury by livestock	13	3%
	Dog fight or bite	5	1%
	Other	10	3%
	Cause not known	5	1%
	Not recorded	1	<0.5%

**Table 8 animals-14-01895-t008:** Non-traumatic illness reported in 444 Australian working farm dogs in a twelve-month period. The types of illnesses and number of dogs that were seen by a veterinarian are shown.

Type of Illness	Number of Dogs (%)	Seen by Veterinarian (%)
	(n = 444)	(n = Dogs with Illness)
Degenerative	18 (4%)	0
Genitourinary	9 (2%)	5 (56%)
Skin	8 (2%)	2 (25%)
Ear or eye	7 (2%)	2 (29%)
Other	33 (7%)	7 (21%)
All types of illness	57 (13%)	15 (26%)

## Data Availability

The datasets used and/or analysed during the current study are available from the corresponding author on reasonable request.

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
