# Peer review of "Owner-Reported Health Events in Australian Farm Working Dogs"

_animals, 2024, doi:10.3390/ani14131895_

Round 1

Reviewer 1 Report

Comments and Suggestions for Authors

The authors conducted a descriptive questionnaire-based study to assess the overall health of working farm dogs, a topic that has received little previous attention in Australia. Questionnaire topics included preventative health care, prevalence of health events (traumatic injuries and non-traumatic illnesses), and factors associated with dogs retiring, dying, or being euthanized. Given the health and welfare focus of this research, the following paper might be of interest to include: Littlewood and Mellor. 2016. Changes in the welfare of an injured working farm dog assessed using the Five Domains Model. Animals 6, 58; doi:10.3390/ani6090058. 

My specific comments for each section are detailed below.

Simple Summary:

Lines 11-12: The wording makes it sound like the dogs reported an illness or injury; editing is  needed.

Lines 14-15: This sentence on study aims should go right after the first sentence of the Simple Summary, before results are reported.

Abstract:

Lines 26-27: Same comment as above about moving study aims to right after first sentence.

Key words: “Working farm dog”, “working dogs” and “Australia” are all in the title, so it would be better to replace these key words with others not in the title (e.g., “questionnaire”; “health event”; “veterinary care”).

Introduction:

Lines 55-58: Study aims are clearly stated.

Materials and methods:

Lines 64-65: The text states that “Data were collected through face-to-face interviews…”. Does this mean that individual researchers asked the questions and completed the questionnaires based on owner answers or did the dog owners complete the questionnaire? If the latter, then the word “interview” seems incorrect because it suggests a back-and-forth interaction between owner and researcher, and should be changed here and in the Simple Summary (line 8) and Abstract (line 20). Approximately how long did the questionnaire take to complete? This information is typically provide in questionnaire studies and it also relates to your Discussion section comparing your methods of data collection (e.g., at a busy event at which owners could not prepare in advance) with those in the New Zealand studies (lines 240-243). These details will help readers better understand your methods of data collection.

Line 90: Thank you for including the full questionnaire – this will make it easier for readers to understand your study and for other researchers to replicate it. Could you make the questionnaire  available as an Appendix, instead of as Supplementary Material, so it is immediately available to readers (i.e., no clicks needed)?

Line 115: Call-out for Table 1 is needed here, although Table 1 might not be necessary, given the full questionnaire is available with the same information.

Results:

Tables are clear and nicely organized.

Line 175: Check error message here and throughout Results section.

Discussion:

Line 233: How does the median age of death of dogs in your study (8 years) compare to age of death for non-working kelpies, dogs of similar size, or working dogs generally?

Lines 302-303: There is a substantial literature on the costs and benefits of neutering for dogs that you might want to discuss/reference here (e.g., Hart et al. 2014. Long-term health effects of neutering dogs: Comparison of Labrador Retrievers with Golden Retrievers. PLOS ONE July 2014, Volume 9, Issue 7, e102241; Also, see discussion in D’Onise K, Hazel S, Caraguel C. Mandatory desexing of dogs: one step in the right direction to reduce the risk of dog bite?

A systematic review. Injury Prevention 2017, 23, 212–218.)

Typically, there is a separate paragraph devoted to study limitations. Here, however limitations are brought up in various places (e.g., 129-132; 240-250; 284-288). The limitations might be better understood by readers if they were grouped together in a single paragraph in the Discussion.

Institutional Review Board Statement:

Lines 335-339: Does “peer review” mean by members of the IRB or some other administrative body? Please specify. Also, if owners were not “interviewed” as questioned above in regard to lines 64-65, then change “dog owners were free to decline to be interviewed” to “dog owners were free to decline to complete the questionnaire.” 

Reviewer 2 Report

Comments and Suggestions for Authors

Review of Pattison, Isaksen, and Cogger’s Owner-reported health events in Australian farm working dogs, manuscript ID animals-3075267.  Pattison et al. describe the results of a study done in 2015 on the health of dogs working on farms in Australia.  While the manuscript is well written, its purely descriptive nature along with the author’s argument that countries with different environments and other factors may have different health outcomes for working dogs limit the generalizability and interest in the results.

Line 51:  In justifying the importance of the current research, it might be helpful to more fully describe the differences in the environments, etc. in New Zealand and Australia.  This would help the reader to understand why you expect the results to be different for the two countries and why the manuscript is important.

Line 61:  The data were collected nine years ago.  Much has happened in the world during those nine years – a pandemic and possibly different economic conditions to name a couple that might impact your findings if the study was done again today.  Perhaps you could include a brief discussion of this possible limitation in the discussion section.

Minor issues:

Line 36:  o should be a degree symbol °.

Lines 115, 161, 162, 176: “Error! Reference source not found” should be fixed.

Lines 118-120 are awkward.

Table 6, Herding dogs trials.  The frequencies sum to 422 (47 + 36 + 4 + 332 + 3) but the table title states 412 dogs.  The categories appear to be mutually exclusive of each other.

 Line 238:  Insert “Zealand” after “New”.

Comments on the Quality of English Language

A few very minor corrections are needed.
